# Molecularly Imprinted Nanoparticles (NanoMIPs) Selective for Proteins: Optimization of a Protocol for Solid-Phase Synthesis Using Automatic Chemical Reactor

**DOI:** 10.3390/polym13030314

**Published:** 2021-01-20

**Authors:** César Cáceres, Ewa Moczko, Itsaso Basozabal, Antonio Guerreiro, Sergey Piletsky

**Affiliations:** 1Departamento de Polímeros, Facultad de Ciencias Químicas, Universidad de Concepción, 4070371 Concepción, Chile; cecaceres@udec.cl; 2Facultad de Ingeniería y Ciencias, Universidad Adolfo Ibáñez, 2562307 Viña del Mar, Chile; 3Department of Chemistry, University of Leicester, Leicester LE1 7RH, UK; ibmoyua@gmail.com (I.B.); ag482@leicester.ac.uk (A.G.); sp523@leicester.ac.uk (S.P.); 4MIP Diagnostics, Leicester MK44 1LQ, UK

**Keywords:** molecularly imprinted nanoparticles, solid-phase synthesis, automatic chemical reactor, protein imprinting

## Abstract

Molecularly imprinted polymer nanoparticles (nanoMIPs) are receiving broad interest as robust and highly selective synthetic receptors for a variety of molecules. Due to their stability, inexpensive synthesis and easy implementation, they are considered a promising alternative to antibodies in sensors, diagnostics and separation applications. The most challenging targets for the production of synthetic receptors are proteins due to their fragile nature and the multitude of possible binding sites in their structure. Herein, we describe the modification and optimization of the protocol for synthesis of nanoMIPs with specificity for proteins using the prototype of an automated solid-phase synthesizer. Using an automated system gives an advantage for the simple, fast and fully controlled, reproducible production of nanoMIPs. The molecular imprinting in the reactor is performed using a template covalently immobilized on a solid support, in mild conditions suitable for preserving protein native structure. The validation of the protocol was made by assessing the ability to regenerate a solid-phase, and by measuring affinity and specificity of nanoparticles. As a model protein, we have chosen trypsin since its enzymatic activity can be easily monitored by using a commercial colorimetric assay. Different protocols were tested for their ability to improve the yield of high affinity nanoparticles in the final elution.

## 1. Introduction

Selective detection and quantification of proteins are crucial in a wide variety of fields, including the pharmaceutical industry, clinical diagnostics, therapeutic monitoring and biotechnology. Protein recognition is based on their selective binding by antibodies as the most suitable biological receptors. Standard tests, which are used in analytical laboratories to detect and quantify proteins in biological samples, are immunoassays [1]. Although antibodies meet the requirements for high specificity and selectivity, they demand a costly, tedious and time-consuming procedure for efficient screening and production [2,3]. Moreover, they suffer from several fundamental limitations, e.g., relatively low chemical and physical stability and often a high price. Therefore, alternative synthetic receptors such as protein-imprinted polymers can offer significant advantages over natural biological counterparts, including high mechanical/chemical stability, ease of preparation, potential re-usability, and low manufacturing cost. Molecularly imprinted polymers (MIPs) have demonstrated the potential to revolutionize the future technology of molecular recognition [4,5,6,7,8]. They have gained a considerable reputation as a cost-effective alternative to bioreceptors in a variety of applications [9,10,11]. Recently, a conventional method for preparing MIPs by bulk co-polymerization and subsequent grinding of polymer monoliths has been replaced by the synthesis of micro- and nanoparticles [2,12,13]. New formats of MIPs offer better control over the quality and reproducibility of polymer synthesis, and higher affinity and specificity over the target molecules [9,14].

This manuscript provides a comparative analysis of several protocols applied in molecular imprinting of proteins using an automatic synthesizer. Generally, imprinting is performed in conditions that have minimal impact on the native structure of the proteins. Therefore, these protocols were tested for the ability to regenerate a solid-phase with trypsin as an immobilized template. Trypsin has been selected as a model protein as its activity can be easily and rapidly monitored using a standard trypsin colorimetric assay.

The first test included a comparison of a common protocol used in our laboratory over the years [15,16]. The last one is based on using a polymer composition that contains *N*-isopropylacrylamide (NIPAM). This composition allows for the elution of nanoparticles at a low temperature, which in principle can be beneficial for improving the stability of the protein template. The second test included a comparison of the thermo-elution of nanoMIPs with elution triggered by a change in pH. In this case, elution of the particles was carried out at pH 5.0 and pH 8.0. The third test involved testing the effect of the presence of a surfactant to facilitate the elution of high-affinity nanoparticles. All work was performed using the prototype of an automatic synthesizer, which in principle should be suitable for industrial manufacturing of nanoMIPs. The conditioning of the solid phase, injection of the polymerization mixture, synthesis of nanoparticles, and washing and elution of the nanoMIP were all under computer control and required minimal manual intervention [17,18].

## 2. Materials and Methods

### 2.1. Automatic Synthesizer

Synthesis of nanoMIPs was performed using an automated synthesizer (see Figure 1 and Figure 2), which was manufactured by HEL Ltd., Borehamwood, UK. It consisted of a temperature-controlled column reactor packed with glass beads bearing the immobilized template. The column consisted of a sliding lid mechanism, which included all inlets and a stirring system. The internal temperature of the reactor was controlled with a dual heating system and monitored via an internal thermocouple fitted in the lid. A set of pumps delivered the monomer mixture, initiator, additional solutions for post-derivatization, and washing and elution solvents, while on the outlet, a fraction collector separated waste streams from high-affinity product fractions. It is also possible to perform post-derivatization of nanoMIPs with different compounds in a solution. The machine also included an N_2_ inlet to flush the reactor before the polymerization and to force out the liquid and empty the reactor under positive pressure. All the parameters and components of the reactor were controlled by a computer and a suitable software (WinISO) and were able to be programmed in advance by the operator.

### 2.2. Preparation of Glass Beads with Immobilized Template

Initially, the solid-phase (glass beads with an average diameter of 90 μm, Spheriglass A, Potters, UK) was conditioned and prepared for further synthesis. This required activation of glass beads with 1 mol L ^−1^ sodium hydroxide and then silanization with 2% *v/v* solution of 3-(aminopropyl)trimethoxysilane in dry toluene overnight to obtain -NH_2_- bearing beads, which allowed further immobilization of target molecules through the suitable linker. Then, the beads were washed with acetone and rinsed with Milli Q water. The template (0.5 mg mL^−1^ of trypsin) was activated in PBS buffer adjusted to pH 6, by using EDC (*N*-(3-dimethylaminopropyl)-*N*-ethylcarbodiimide) and NHS (*N*-hydroxysuccinimide) at 10 and 15 mg/mL, respectively, for 15 min. The solution was then adjusted to pH 7.5 with 0.1 M sodium hydroxide, added to the beads and left for overnight incubation. All chemicals and solvents used in the solid-phase preparation, and the synthesis of nanoMIPs was purchased from Sigma-Aldrich (UK) without further purification.

### 2.3. Synthesis of NanoMIPs with Elution of High Affinity Nanoparticles at High Temperature (Standard Protocol)

The procedure of the synthesis of nanoMIPs in water was adapted from Hoshino et al. [19]. In 100 mL of Milli Q water, the following monomers were dissolved: 39 mg of *N*-isopropylacrylamide (NIPAm), 2 mg of *N*,*N*′-methylenebisacrylamide (BIS), 33 mg of *N*-tert-butylacrylamide (TBAm) and 2.2 µL of acrylic acid (AA). TBAm was first dissolved in 2 mL EtOH and then added to the aqueous solution. The solution was degassed under vacuum and sonicated for 10 min, and then purged with N_2_ for 30 min. The polymerization was initiated with 800 µL of ammonium persulfate aqueous solution (APS) and 30 µL of *N*,*N*,*N*′,*N*′-tetramethylethylenediamine (TEMED), and performed over different time periods at ambient temperature. The elution of high affinity nanoMIPs was performed five times using 20 mL of Milli Q water at 60 °C. The synthesis was performed five consecutive times using the same glass beads.

### 2.4. Synthesis of NanoMIPs with Elution of High Affinity Nanoparticles at Low Temperature

The polymerization mixture was prepared by mixing 0.452 g NIPAm, 0.034 g BIS, 56 mg APS and 4.9 TEMED. All compounds were dissolved in water. TBAm was first dissolved in 2 mL EtOH and then added to the aqueous solution. The solution was degassed under vacuum and sonicated for 10 min, and then purged with N_2_ for 30 min. The polymerization was performed in a water bath at 37 °C for 90 min. The washing step was carried out using 14 times 20 mL of Milli Q water at 37 °C. The elution of high affinity nanoMIPs was performed five times in 20 mL of Milli Q water at 20 °C. The synthesis was performed five consecutive times using the same glass beads.

### 2.5. Evaluation of the Solid-Phase Stability

The enzymatic activity of the trypsin immobilized onto the solid-phase was assessed by using the Trypsin Activity Assay Kit (Sigma-Aldrich) pre- and post-nano MIPs synthesis. The assay was performed to determine its activity using Nα-benzoyl-L-arginine 4-nitroanilide hydrochloride (Bz-Arg-pNA·HCl) as the substrate. The procedure was as follows: 2 mL of the solution containing the substrate (10 mg of Bz-Arg-pNA·HCl, 2 mL of dimethyl sulfoxide and 15 mL TRIS buffer) was added to 0.5 g glass beads with trypsin immobilized onto the surface. After 2 and 24 h the supernatant was collected and its optical absorbance measured at 405 nm. The results were compared with the absorbance of corresponding standard solutions. The synthesis was performed five consecutive times using the same glass beads.

### 2.6. Elution at Different pHs

Synthesis of nanoMIPs for this experiment was performed as described above in Section 2.4. The polymerization of the nanoMIPs and the subsequent washing steps were performed at ambient temperature. Elution of high-affinity nanoparticles was performed at 4 °C using 100 mM sodium acetate buffer, pH 5.0, and 100 mM and sodium phosphate buffer, pH 8.0.

### 2.7. Elution Using Surfactant

In this experiment, synthesis of nanoMIPs was performed as described in Section 2.6. The polymerization of the nanoMIPs and the subsequent washing steps were performed at ambient temperature. Elution of high-affinity nanoparticles was performed at 4 °C using 100 mM sodium acetate buffer, pH 5.0 and 100 mM and sodium phosphate buffer, pH 8.0 containing Tween 20.

### 2.8. Analysis of NanoMIPs by Dynamic Light Scattering (DLS)

To verify the size of the synthesized nanoparticles, the eluted fractions were analyzed using a Zetasizer Nano (Nano-S) from Malvern Instruments Ltd. (Malvern, UK). Solutions of nanoMIPs in water (1 mL) were sonicated for 10 min, filtered through glass fiber syringe filters (pore size 1.2 μm pore size) and analyzed by DLS at 25 °C in a 3 cm^3^ disposable polystyrene cuvette.

### 2.9. Analysis of NanoMIPs by Transmission Electron Microscopy (TEM)

The TEM images were obtained using the Philips CM20 transmission electron microscope. Samples were sonicated for 1 min, and then a drop of the sample was placed on a carbon-coated copper grid and dried in air.

### 2.10. Analysis of NanoMIPs Affinity by Biacore

Affinity analysis of nanoMIPs was performed with a BIAcore 3000 SPR system (BIAcore, Sweden) using gold chips with the templates immobilized on the surface. Kinetic data were fitted using BIAEvaluation Software v4.1 (BIAcore, Sweden).

## 3. Results and Discussion

The high affinity nanoMIPs were produced for trypsin as a model template in order to optimized protocol of an automatic imprinting of proteins by using the new reactor. The activity of trypsin immobilized in the glass beads was measured after 2 h of the reaction. In addition, a comparison between nanoparticles synthesized with native trypsin and nanoparticles synthesized with denatured trypsin was carried out. This study intended to prove whether denaturing of protein as a template in the process of MIP preparation was crucial to the recognition of native proteins. The yield and the affinity analyses were performed for both cases.

The production of a trypsin solid-phase was achieved using a known protocol for the immobilization of templates containing primary amino groups [15,20,21]. The activation of the glass bead surface with NaOH can increase the amount of reactive OH groups present on the surface prior to salinization, which then promotes reactivity. The mild conditions of chemically-initiated aqueous synthesis described in this work are particularly compatible with the synthesis of nanoMIPs for large templates, such as enzymes, antibodies and other biological molecules [16].

First experiments included cold elution with thermo-responsive polymer nanoparticles. Synthesis at ambient temperature, followed by cooling the reactor, was particularly beneficial for protein imprinting. Elution under these conditions was based on the reversible temperature-induced swelling of the polymer at 4 °C, due to the presence of *N*-isopropylacrylamide in the polymer composition [10,22]. Polymers containing this monomer can undergo a reversible phase transition from collapsed to swollen by a decrease in temperature. The swelling resulted in the distortion of the binding site of the MIP nanoparticles with a consequent detachment of the particle from an immobilized template. The subsequent warming/re-heating of the nanoparticles at ambient temperature restored their recognition properties. The results of testing this approach in the production of nanoMIPs for trypsin are summarized in Table 1 and Table 2.

As can be seen in Table 1, immobilized trypsin was losing activity after each cycle. Only 10% of enzyme activity remained after five cycles of nanoMIPs preparation. It is essential to mention that the reverse cooling-heating process even in the absence of monomer mixture led to decreased enzyme activity. It can also be seen that the yield of nanoparticles decreased after each cycle (see Table 2). The yield and size of the nanoMIPs were analyzed separately after each cycle. Thus, it can be concluded that the approach adopted here did not have additional benefits in terms of the activity of the template and yield of nanoparticles.

Considering all the experiments involving the elution of high affinity of nanoMIPs) for trypsin at different pHs, two different pHs were used in order to elute high-affinity nanoparticles: acidic and basic (pH 5.0 and pH 8.0). Results are summarized in the following Tables (Table 3 and Table 4). The yield and size of produced nanoMIPs are presented in Table 5 and Table 6, respectively.

It seemed that changes in pH offered more significant results than changes in temperature. In these conditions, the yield of particles decreased by half after five cycles and the activity of immobilized enzymes was better preserved. No significant difference in the yield of particles was observed by applying an acidic or basic wash; however, the activity of immobilized enzyme decreased more significantly during basic conditions.

Further experiments included using a surfactant (Tween 20) in the elution of nanoMIPs. It was expected that the surfactant may improve the elution of high-affinity nanoparticles. Results of the measurements are summarized in the following tables (Table 7 and Table 8).

After the five successive syntheses, the remaining activity of trypsin immobilized on the glass beads was only 17%. The same activity was detected for immobilized enzyme treated in identical conditions but in the absence of the polymerization mixture. The yield of high-affinity nanoparticles after five cycles was lower than for other processes tested in the studies. It might be concluded that the inability to re-cycle protein template lies not in the polymerization procedure or elution conditions but in the continuous denaturation of the template itself.

Finally, the nanoMIPs synthesized in the presence of native and denatured protein were compared. In order to denature the protein, the glass beads containing trypsin as template were boiled in water for 30 min. Subsequently, the polymerization mixture was added, polymerization was performed and particles were eluted at a low temperature (4 °C), as shown in the first experiment. As in the previous case, particles were collected after each re-cycling stage. The results describing the yield and size of nanoparticles are summarized in Table 9.

The yield of nanoparticles made for the denatured template was ten times lower after five cycles when compared with the first cycle, as shown in the first results. This result indicates that the decrease in the yield of nanoparticles after each successive cycle was not related to the continued denaturing of the protein template. Similar results were observed regarding their size and polydispersity index. However, based on that data, a definitive conclusion cannot be made (no clear pattern for each re-cycling stage is observed), but the results obtained on the yield indicate a low efficiency in reusing protein as a template. The typical TEM image of nanoMIPs is presented in Figure 3. The results again prove that reuse of the solid-phase for the proteins has its issues, but the problem can potentially be solved by immobilizing and imprinting the protein epitope [23].

Finally, affinity analysis, the interaction analysis was performed using a Biacore 3000 instrument at 25 °C using PBS as the running buffer at a flow rate of 15 µL min^−1^. The aqueous suspensions of the tested nanoparticles were diluted in PBS for the analysis following the series of 2× dilutions. The range of concentrations of non-modified nanoMIPs was from 0.12 nmol to 0.97 nmol L^−1^.

Dissociation constants (KD) were calculated from plots of the equilibrium biosensor response using the BiaEvaluation v4.1 software using a 1:1 binding model with drifting baseline fitting. A dissociation constant of the interactions between native trypsin-specific nanoMIPs and trypsin immobilized on the surface of the Biacore chip was estimated as 9 × 10^−9^ nmol L^−1^. On the other hand, a dissociation constant value of 3.39 × 10^−9^ nmol L^−1^ was estimated between the denatured trypsin nanoparticles and trypsin immobilized on the chip. It is interesting that nanoparticles prepared for denaturing trypsin are capable of strong binding to native trypsin immobilized on a Biacore chip. One possible explanation to this is the possibility that protein immobilized onto the flat gold surface has its structure partly denatured. More work would be required to test recognition properties of MIP nanoparticles in solution. The results are presented in Figure 4 and Figure 5. They show SPR sensorgrams for native trypsin-specific nanoMIPs and denatured trypsin-specific nanoMIPs, respectively.

## 4. Conclusions

To summarise this work, the best results in terms of yield of nanoparticles after five cycles of polymerization were obtained in the protocol which relied on change of pH in elution of nanoMIPs. The use of surfactant provided quite mild conditions for preserving activity of immobilized trypsin. However, all treatments tested in this work were ineffective for regeneration of solid-phase with immobilized protein. The yield of nanoparticles diminished after each subsequent re-cycling step, as along with the activity of immobilized enzyme. We can conclude that the imprinting of whole proteins can be recommended only for research purposes. All large-scale application of protein-imprinted nanoMIPs will have to be considered only for particles prepared using epitope imprinting.

The nanoMIPs were synthesized for trypsin and collected using different elution strategies. Cold elution using thermo-responsive nanoparticles (taking into account the presence of *N*-isopropylacrylamide in the polymer composition) was performed, obtaining particles with a yield 0.145 mg mL^−1^. Immobilized trypsin in these conditions preserved 10% of its activity after five cycles. By using different pHs in the elution step, the results were very similar. Trypsin activity decreased remarkably, and the yield of nanoparticles was 0.118 and 0.249 mg mL^−1^ for pH 5.0 and 8.0, respectively. In addition, when surfactant was used, the yield of high-affinity nanoMIPs was 0.439 mg mL^−1^. The remaining trypsin activity was 17%.

The above results have demonstrated that after five cycles of synthesis of nanoparticles, the protein lost most of its activity. The yield of nanoparticles was also reduced after each manufacturing cycle. This phenomenon, however, is not related with denaturing of the enzyme molecule since the same effect can be seen for already denatured enzyme. It can be concluded that the solid phase with trypsin immobilized on the surface can only be used for one to two cycles, due to low remaining activity of the enzyme. It is expected that epitopes will represent much better targets for creating protein-specific MIPs. Interestingly, nanoMIPs synthesized with native and denatured trypsin have similar affinity to native trypsin immobilized on Biacore chip. The significance of this observation is under investigation.

## Figures and Tables

**Figure 1 polymers-13-00314-f001:**
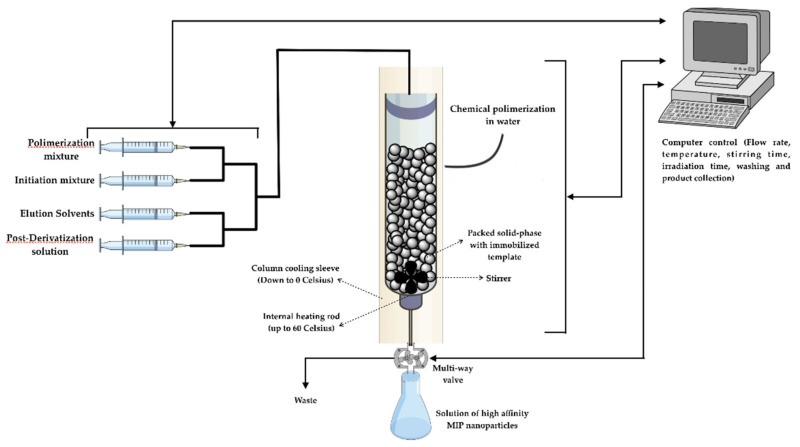
Schematic illustration of the automatic reactor.

**Figure 2 polymers-13-00314-f002:**
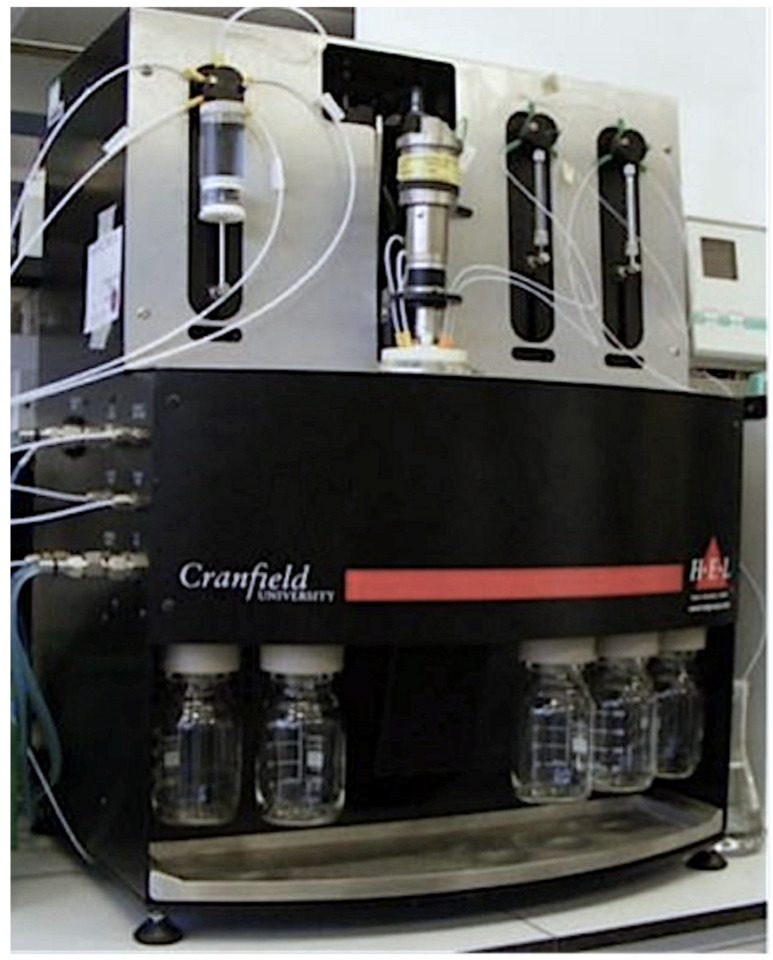
Photography of the automatic chemical reactor.

**Figure 3 polymers-13-00314-f003:**
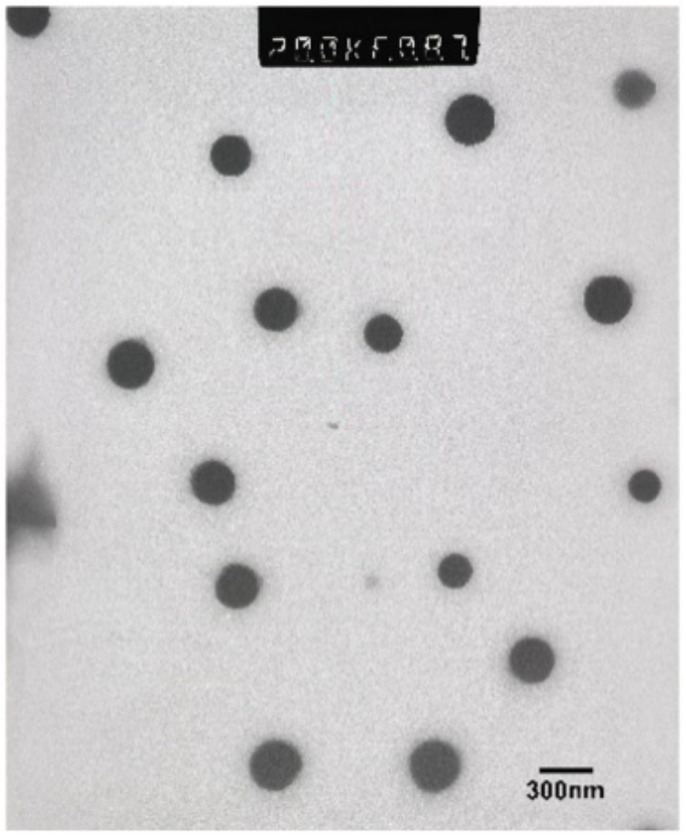
TEM image of dried nanoMIPs synthesized for native protein.

**Figure 4 polymers-13-00314-f004:**
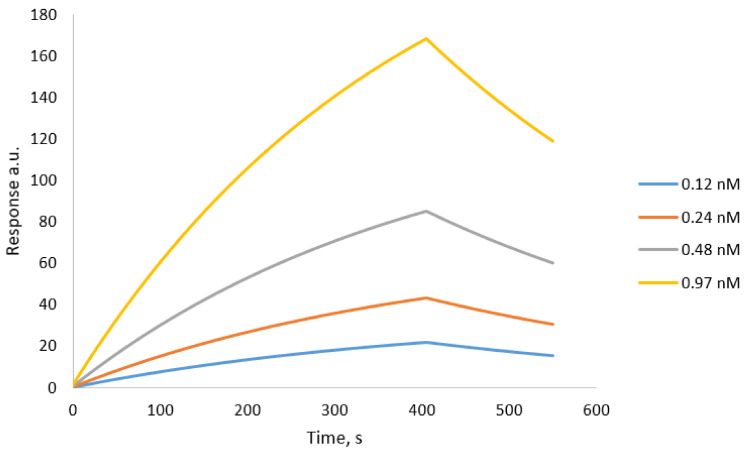
SPR sensorgrams for native trypsin-specific nanoMIPs. They were injected onto specific trypsin-coated sensor surface. Solutions of NPs were injected at concentrations ranging from 0.12 nM to 0.97 nM. SPR tests were performed in PBS buffer pH 7.4 at 25 °C.

**Figure 5 polymers-13-00314-f005:**
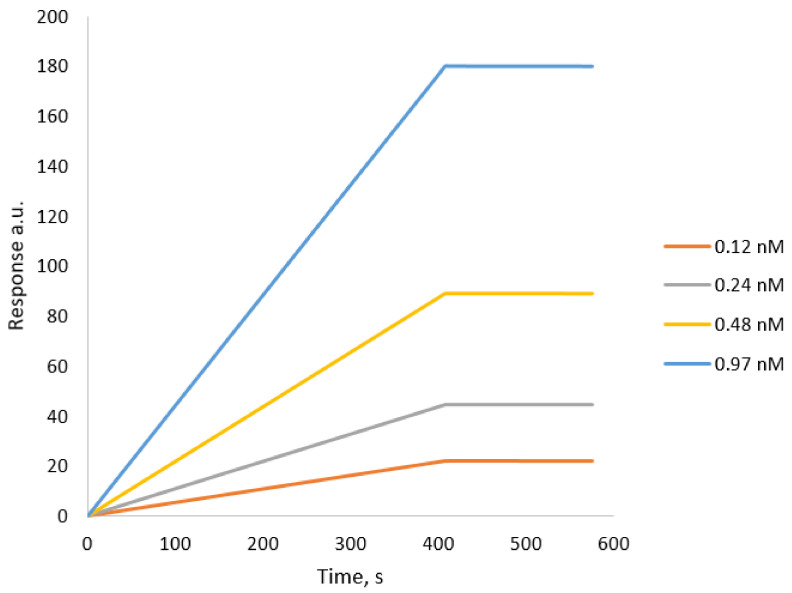
SPR sensorgrams for denatured trypsin-specific nanoMIPs. They were injected on specific trypsin-coated sensor surface. Solutions of NPs were injected at concentrations ranging from 0.12 nmol L^−1^ to 0.97 nmol L^−1^. SPR tests were performed in PBS buffer pH 7.4 at 25 °C.

**Table 1 polymers-13-00314-t001:** Trypsin activity after successive synthesis cycles. The activity was measured after 2 h at 405 nm.

CYCLES	Remaining Activity
1	50%
2	33%
3	25%
4	18%
5	11%
Control	20%

**Table 2 polymers-13-00314-t002:** Yield and size of molecularly imprinted polymers (MIP) nanoparticles (mg mL^−1^) after five successive cycles. The diameter of the particles and polydispersity index value (PDI).

CYCLES	Concentration (mg mL^−1^)	Diameter (nm)	PDI (nm)
1	0.145	257 ± 62.6	0.339 ± 0.1
2	0.054	124 ± 16.9	0.190 ± 0.2
3	0.045	109 ± 17.0	0.155 ± 0.1
4	0.039	155 ± 15.1	0.599 ± 0.3
5	0.019	149 ± 15.2	0.466 ± 0.2

**Table 3 polymers-13-00314-t003:** Trypsin activity after successive synthesis cycles. Elution of high-affinity MIP nanoparticles was performed at pH 5.0. The activity was measured after 2 h at 405 nm.

CYCLES	Remaining Activity
1	59%
2	48%
3	26%
4	21%
5	14%
Control	43%

**Table 4 polymers-13-00314-t004:** Trypsin activity after successive synthesis cycles. Elution of high-affinity MIP nanoparticles was performed at pH 8.0. The activity was measured after 2 h at 405 nm.

CYCLES	Percentage of Activity
1	21%
2	11%
3	11%
4	6.4%
5	10%
Control	29%

**Table 5 polymers-13-00314-t005:** Yield and size of MIP nanoparticles after five successive cycles and elution at pH 5.0. For the measurements of size PDIs were included.

CYCLES	Concentration (mg mL^−1^)	Diameter (nm)	PDI (nm)
1	0.118	81.5 ± 2.6	0.136 ± 0.1
2	0.122	150.2 ± 26.6	0.345 ± 0.3
3	0.094	136.0 ± 45.0	0.213 ± 0.1
4	0.089	143.2 ± 14.6	0.326 ± 0.1
5	0.096	167.3 ± 39.0	0.421 ± 0.2

**Table 6 polymers-13-00314-t006:** Yield and size of MIP nanoparticles after five successive cycles and elution at pH 8.0. For the measurements of size PDIs were included.

CYCLES	Concentration (mg mL^−1^)	Diameter (nm)	PDI (nm)
1	0.249	170.0 ± 13.9	0.200 ± 0.2
2	0.347	147.0 ± 14.2	0.152 ± 0.1
3	0.130	132.0 ± 4.9	0.114 ± 0.1
4	0.118	118.0 ± 5.9	0.184 ± 0.2
5	0.126	130.0 ± 1.7	0.114 ± 0.3

**Table 7 polymers-13-00314-t007:** Trypsin activity after successive synthesis cycles and elution using Tween 20. The activity was measured after 2 h at 405 nm.

CYCLES	Remaining Activity
1	56%
2	56%
3	33%
4	41%
5	17%
Control	17%

**Table 8 polymers-13-00314-t008:** Yield and size of MIP nanoparticles after five successive cycles and elution using Tween 20. For the measurements of size, PDIs were included.

CYCLES	Concentration (mg mL^−1^)	Diameter (nm)	PDI (nm)
1	0.439	303.3 ± 3.8	0.242 ± 0.1
2	0.103	218.5 ± 35.2	0.558 ± 0.1
3	0.125	244.2 ± 14.5	0.366 ± 0.2
4	0.112	178.8 ± 7.43	0.425 ± 0.1
5	0.061	221.2 ± 22.5	0.328 ± 0.1

**Table 9 polymers-13-00314-t009:** The concentration of MIP nanoparticles after different cycles using denatured trypsin in the solid-phase.

CYCLES	Concentration (mg mL^−1^)	Diameter (nm)	PDI (nm)
1	0.342	297.0 ± 32.24	0.239 ± 0.2
2	0.148	154.0 ± 16.93	0.230 ± 0.2
3	0.072	215.0 ± 17.01	0.245 ± 0.4
4	0.023	145.0 ± 45.10	0.699 ± 0.3
5	0.028	155.0 ± 30.17	0.536 ± 0.3

## Data Availability

Not applicable.

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
