# Peer review of "Molecularly Imprinted Nanoparticles (NanoMIPs) Selective for Proteins: Optimization of a Protocol for Solid-Phase Synthesis Using Automatic Chemical Reactor"

_polymers, 2021, doi:10.3390/polym13030314_

Round 1
Reviewer 1 Report
-The English of the manuscript must be improved. The authors are encouraged to ask a native English speaker to proofread the manuscript.
-Lines 90-91: Please provide an explanation for the role of succinic anhydride solution in section 2.2.
-Line 92-92: Throughout the manuscript, all the information for the used materials (e.g. venders …) and methods must be mentioned. In “0.5 mg ml-1 of trypsin”, what is the solvent? Did you buy it or prepare it in PBS buffer pH 6? Please provide more detailed information for the used procedure. The same for EDC and NHS…
-Line 105: In “and allowed to run for”, what do you mean by run?
-Line 120: What does “substrate“ mean? A solution? Please use the right words.
-Line 139: In “1.2 mm glass fibre syringe filter“, what is the 1.2 mm? What is the pore size?
-Line 151-152: How was the activity of trypsin measured after 2 hours of the reaction? In section 2.5., a method is described for the stability evaluation of the trypsin immobilized onto solid-phase. Did authors measure the stability and reported as activity? If this is true, then i) please use the same terms clearly throughout the manuscript and ii) how did the stability or remained trypsin was measured between cycles? Please provide more information or suitable references for section 2.5. explaining the happening reactions.
-Line 163: “First experiments included cold elution…”, However, synthesis of polymers with elution at high temperature is the first one which is described in experimental section. Please present the result as the order described in previous sections.
Table 2: Does each cycle collected separately or each new cycle is added to the previous one? Please add this information. Why is the diameter in the first cycle is bigger that the other cycles?
-Line 232: What is nm in “0.12 nm”?
Author Response
Response to Reviewer
We really appreciate your time and the effort to revised enclosed manuscript. All
suggested corrections were incorporated into the text. Please find enclosed the revised
version of the manuscript and below the detailed response regarding your comments.
Comment 1: The English of the manuscript must be improved. The authors are encouraged to
ask a native English speaker to proofread the manuscript.
Answer 1: The English of the manuscript has been corrected.
Comment 2: Lines 90-91: Please provide an explanation for the role of succinic anhydride
solution in section 2.2.
Answer 2: There was a mistake in the manuscript, we are very sorry for that and really
appreciate the remark. The text has been corrected: Initially the solid-phase (glass beads with
average diameter of 90 μm, Spheriglass A, Potters, UK) were conditioned and prepared for
further synthesis. This required activation of glass beads with 1 mol L -1 sodium hydroxide and
then silanization with 2 % v/v solution of 3-(aminopropyl)trimethoxysilane in dry toluene overnight
to obtain -NH2- bearing beads, which allowed further immobilization ation of target molecules
through the suitable linker. Then the beads were washed with acetone and rinsed with Milli Q
water. The template (0.5 mg ml-1 of trypsin) was activated in PBS buffer adjusted to pH 6, by
using EDC (N-(3-dimethylaminopropyl)-N-ethylcarbodiimide) and NHS (N-hydroxysuccinimide) at
10 and 15 mg/mL respectively for 15 min. The solution was then adjusted to 7.5 with 0.1 M
sodium hydroxide, added to the beads and left for overnight incubation.
Comment 3: Line 92-92: Throughout the manuscript, all the information for the used materials
(e.g. venders …) and methods must be mentioned. In “0.5 mg ml-1 of trypsin”, what is the
solvent? Did you buy it or prepare it in PBS buffer pH 6? Please provide more detailed
information for the used procedure. The same for EDC and NHS…
Answer 3: All chemicals, solvents and most of the materials were purchased from Sigma-Aldrich
(UK). This information has been added into the text.
Comment 4: Line 105: In “and allowed to run for”, what do you mean by run?
Answer 4: It means that the polymerization was performed over different time periods. The
information has been corrected. Thank you.
Comment 5: Line 120: What does “substrate“ mean? A solution? Please use the right words.
Answer 5: The explanation has been given in the text. “The enzymatic activity of the trypsin
immobilized onto the solid-phase was assessed by using the Trypsin Activity Assay Kit (Sigma-
Aldrich) prior and post nanoMIPs synthesis. The assay was performed in order to determine its
activity using Nα-benzoyl-L-arginine 4-nitroanilide hydrochloride (Bz-Arg-pNAØHCl) as the
substrate. The procedure was as follows: 2 mL of the solution containing the substrate (10 mg of
Bz-Arg-pNAØHCl, 2 mL of dimethyl sulfoxide and 15 mL TRIS buffer) was added to 0.5 g glass
beads with trypsin immobilized onto the surface. After 2 and 24 hours the supernatant was
collected, and its optical absorbance measured at 405 nm. The results were compared with
absorbance of corresponding standard solutions. The synthesis was performed 5 consecutive
times using the same glass beads.”
Comment 6: Line 139: In “1.2 mm glass fibre syringe filter“, what is the 1.2 mm? What is the
pore size?
Answer 6: The information has been corrected. “Solutions of nanoMIPs in water (1 mL) were
sonicated for 10 min, filtered through glass fibre syringe filters (pore size 1.2 μm) and analysed by
DLS at 25 oC in a 3 cm3 disposable polystyrene cuvette.
Comment 7: Line 151-152: How was the activity of trypsin measured after 2 hours of the
reaction? In section 2.5., a method is described for the stability evaluation of the trypsin
immobilized onto solid-phase. Did authors measure the stability and reported as activity? If this is
true, then i) please use the same terms clearly throughout the manuscript and ii) how did the
stability or remained trypsin was measured between cycles?
Answer 7: Correct, only enzymatic activity of trypsin was measured. The remaining activity is a
good indication of stability in this case.
Comment 8: Line 163: “First experiments included cold elution…”, However, synthesis of
polymers with elution at high temperature is the first one which is described in experimental
section. Please present the result as the order described in previous sections.
Answer 8: It is indicated in the title of the first experimental section that this part includes
“Synthesis of NanoMIPs with Elution of High Affinity Nanoparticles at High Temperature
(Standard Protocol)”. It was done in order to further explaination how the standard protocol was
modified.
Comment 9: Table 2: Does each cycle collected separately or each new cycle is added to the
previous one? Please add this information. Why is the diameter in the first cycle is bigger that the
other cycles?
Answer 9: The yield and size of the nanoMIPs has been analysed after each cycle separately.
Considering higher standard deviation, perhaps in the first cycle they were produced smaller
nanoparticles which might easier aggregate, generating (apparent) bigger hydrodynamic
diameter.
Comment 10: Line 232: What is nm in “0.12 nm”?
Answer 10: The correct value is nM. The range of concentrations of non-modified nanoMIPs was
from 0.12 nmol L-1 to 0.97 nmol L-1.

Reviewer 2 Report
This manuscript describes an optimization of the protocol for synthesis of nanoMIPs with specificity for proteins using prototype of an automated solid-phase synthesizer. The molecular imprinting in the reactor is performed using template covalently immobilized on the solid support, in mild conditions suitable for preserving protein native structure. Immobilized trypsin under optimal conditions preserved 10 % of its activity after 5 cycles. Therefore, I recommend publication of this manuscript after the following minor revisions.
- The polymerization scheme for the preparation of nanoMIPs should be provided.
- At lines 107 and 255, ‘polymerisation’ should be ‘polymerization’.
- The more detailed conditions for TEM measurement should be added in the section 2.10.
- The DLS profiles should be depicted.
- The discussion on PDI values should be mentioned.
Author Response
Response to Reviewer
We really appreciate your time and the effort to revised enclosed manuscript. All
suggested corrections were incorporated into the text. Please find enclosed the revised
version of the manuscript and below the detailed response regarding your comments.
Responses
Comment 1: At lines 107 and 255, ‘polymerisation’ should be ‘polymerization’.
Answer 1: Thank you, the spelling has been corrected.
Comment 2: The more detailed conditions for TEM measurement should be added in the
section 2.10.
Answer 2: The section describing TEM measurements is included in the Materials and Methods.
Comment 3: The DLS profiles should be depicted.
Answer 3: Unfortunately this data has not been saved at the time when experiment was made.
Comment 4: The discussion on PDI values should be mentioned.
Answer 4: This paragraph has been added to the section of the Conclusions. “Similar results
were observed regarding their size and polydispersity index. However, no clear pattern for each
re-cycling stage is observed and the results obtained on the yield indicate low efficiency in
reusing protein as a template”.
